# Enzyme-Mediated Quenching of the *Pseudomonas* Quinolone Signal (PQS): A Comparison between Naturally Occurring and Engineered PQS-Cleaving Dioxygenases

**DOI:** 10.3390/biom12020170

**Published:** 2022-01-21

**Authors:** Alba Arranz San Martín, Jan Vogel, Sandra C. Wullich, Wim J. Quax, Susanne Fetzner

**Affiliations:** 1Institute for Molecular Microbiology and Biotechnology, University of Münster, Corrensstraße 3, 48149 Münster, Germany; a_arra01@uni-muenster.de (A.A.S.M.); wullich@uni-muenster.de (S.C.W.); 2Department of Chemical and Pharmaceutical Biology, Groningen Research Institute of Pharmacy, University of Groningen, Antonius Deusinglaan 1, 9713 AV Groningen, The Netherlands; j.g.t.vogel@rug.nl (J.V.); w.j.quax@rug.nl (W.J.Q.)

**Keywords:** dioxygenase, *Galleria mellonella*, *Pseudomonas aeruginosa*, *Pseudomonas* quinolone signal, quorum quenching, virulence

## Abstract

The opportunistic pathogen *Pseudomonas aeruginosa* employs quorum sensing to govern the production of many virulence factors. Interference with quorum sensing signaling has therefore been put forward as an attractive approach to disarm this pathogen. Here, we analyzed the quorum quenching properties of natural and engineered (2-alkyl-)3-hydroxy-4(1*H*)-quinolone 2,4-dioxygenases (HQDs) that inactivate the *P. aeruginosa* signal molecule PQS (*Pseudomonas* quinolone signal; 2-heptyl-3-hydroxy-4(1*H*)-quinolone). When added exogenously to *P. aeruginosa* cultures, all HQDs tested significantly reduced the levels of PQS and other alkylquinolone-type secondary metabolites deriving from the biosynthetic pathway, such as the respiratory inhibitor 2-heptyl-4-hydroxyquinoline *N*-oxide. HQDs from *Nocardia farcinica* and *Streptomyces bingchenggensis*, which combine low K_M_ values for PQS with thermal stability and resilience in the presence of *P. aeruginosa* exoproducts, respectively, attenuated production of the virulence factors pyocyanin and pyoverdine. A delay in mortality was observed when *Galleria mellonella* larvae were infected with *P. aeruginosa* suspensions treated with the *S. bingchenggensis* HQD or with inhibitors of alkylquinolone biosynthesis. Our data indicate that quenching of PQS signaling has potential as an anti-virulence strategy; however, an efficient anti-virulence therapy against *P. aeruginosa* likely requires a combination of agents addressing multiple targets.

## 1. Introduction

*Pseudomonas aeruginosa* is a ubiquitous environmental bacterium and opportunistic pathogen associated with a wide range of nosocomial infections. It typically affects immunocompromised and hospitalized individuals and is a major pathogen associated with pulmonary infections, particularly in patients with cystic fibrosis (CF) [1,2]. Infections caused by *P. aeruginosa* are increasingly difficult to treat due to intrinsic and acquired antibiotic resistances and the spread of antibiotic-resistant strains [3].

Key factors contributing to the pathogenesis of *P. aeruginosa* include the production of a plethora of virulence factors, the formation of biofilms, and its metabolic versatility and high adaptability [1]. Many virulence-associated behaviors are regulated via quorum sensing (QS), a cell-to-cell communication mechanism employed by bacteria to sense and collectively respond to cell density and environmental cues. QS relies on the production and detection of diffusible signal molecules termed autoinducers (AI), which are detected by cognate receptor proteins. The AI-receptor complexes typically activate transcription of the gene(s) coding for AI synthesis and trigger population-wide changes in gene expression [4]. In *P. aeruginosa*, the QS network is composed of three interconnected signaling circuits, namely the two *N*-acylhomoserine lactone (AHL) dependent signaling systems *las* and *rhl* and the *pqs* system that relies on particular 2-alkyl-4(1*H*)-quinolones (AQs) as signal molecules [1,5]. Besides 2-heptyl-3-hydroxy-4(1*H*)-quinolone (the *Pseudomonas* quinolone signal, PQS) as a major AQ-type signal of *P. aeruginosa*, its biosynthetic precursor 2-heptyl-4(1*H*)-quinolone (HHQ) also is an AI of the *pqs* system. The *pqs* system is involved, either directly or indirectly, in the regulation of a series of virulence factors, including the redox-active pigment pyocyanin, the siderophore pyoverdine, rhamnolipid biosurfactants, elastase, or hydrogen cyanide [1,6,7]. Besides their regulatory role in *P. aeruginosa* pathogenicity, PQS and HHQ possess additional functionalities. PQS induces the formation of outer membrane vesicles, chelates ferric iron and mediates iron acquisition, activates the oxidative stress response, acts as a stress warning signal, and modulates host immune responses [6,7,8]. PQS and HHQ repress motility in several bacterial species, and HHQ exhibits bacteriostatic activity [9].

Another product of the AQ biosynthetic pathway, 2-heptyl-4-hydroxyquinoline *N*-oxide (HQNO), does not act as a QS signal molecule but is a major virulence factor, which moreover triggers autolysis of *P. aeruginosa*. As a potent inhibitor of the cytochrome *bc*_1_ complex, it strongly induces the formation of reactive oxygen species (ROS), leading to disruption of membrane integrity and autolysis. HQNO-mediated release of DNA promotes biofilm formation and antibiotic tolerance [10].

Given the involvement of microbial QS systems in the regulation of virulence factor production, these represent a potential target for anti-virulence therapies [11]. Interference with QS signaling, also termed quorum quenching (QQ), may be achieved using small molecules that inhibit the biosynthesis, detection, or transduction of the signal molecule, as well as through the use of QQ enzymes that catalyze the inactivation of the signal molecules [12]. Numerous QQ enzymes that target AHL signal molecules have been described in the literature, whereas the number of enzymes known to inactivate other types of signal molecules is limited [13].

Within the huge superfamily of α/β-hydrolase fold proteins, the subfamily of 3-hydroxy-4(1*H*)-quinolone 2,4-dioxygenases (HQDs) comprises cofactor-less enzymes that catalyze the cleavage of the heterocyclic ring of 3-hydroxy-4(1*H*)-quinolones [14] (Figure 1). Among them, HodC (1*H*-3-hydroxy-4-oxoquinaldine 2,4-dioxygenase) from *Arthrobacter* sp. Rue61a was the first enzyme described to cleave PQS into carbon monoxide and *N*-octanoylanthranilic acid [15]. Even though its activity toward PQS is low, as its physiological substrate is 3-hydroxy-2-methyl-4(1*H*)-quinolone (an intermediate in the degradation of 2-methylquinoline), the addition of HodC to *P. aeruginosa* cultures quenched AQ production [15]. In contrast to HodC, the dioxygenases AqdC1 from *Rhodococcus erythropolis* BG43 and AqdC of *Mycobacteroides abscessus* subsp. *abscessus* preferentially cleave PQS [16]. These enzymes are part of a PQS-inducible pathway that converts HHQ to octanoate and anthranilic acid, which can be funneled into central metabolic pathways [17,18].

A main drawback of using HodC or AqdC for potential application as QQ enzymes is their susceptibility to proteolytic degradation or denaturation in adverse physicochemical conditions, respectively [15,16]. To improve the robustness of mycobacterial AqdC, a recent study used computational library design to predict stabilizing amino acid replacements and to engineer more thermostable variants [19].

Further members of the HQD family active toward PQS have been recently identified [14]. Among those, HQD*_N.f._* from *Nocardia farcinica* and HQD*_S.b._* from *Streptomyces*
*bingchenggensis* showed higher affinity and catalytic efficiency toward PQS than AqdC and rhodococcal AqdC1, making them promising candidates for QQ studies. In the present study, we compared the QQ properties of two engineered AqdC proteins and the newly described PQS dioxygenases HQD*_N.f._* and HQD*_S.b._* to those of wild-type AqdC. To this end, we analyzed the AQ levels and production of the QS-regulated virulence factors pyocyanin, pyoverdine, rhamnolipids, and elastase by *P. aeruginosa* in the presence of the QQ enzymes. Additionally, the virulence of *P. aeruginosa* incubated with the PQS dioxygenases was assessed in a *Galleria mellonella* infection model that enables the precise injection of defined bacterial loads directly into the hemocoel. For comparison, synthetic inhibitors targeting PqsD, a key enzyme of AQ biosynthesis, were included in the *G. mellonella* studies.

## 2. Materials and Methods

### 2.1. Chemicals

HHQ, HQNO, and PQS were acquired from Sigma Aldrich (Schnelldorf, Germany). PqsD inhibitors (compounds 19 and 62) [20,21] were kindly provided by Dr. Empting and Prof. Hartmann, Helmholtz Institute for Pharmaceutical Research Saarland (HIPS), Saarbrücken, Germany.

### 2.2. Bacterial Strains, HQD Enzymes, and Plasmids

HQDs were recombinantly produced in *E. coli* BL21(DE3) [22] or *E. coli* TOP10 (Invitrogen). The proteins, source organisms, and expression plasmids used are listed in Table 1. Assays to determine quenching of virulence factor production by PQS dioxygenases were performed with *P. aeruginosa* strains PAO1 (Nottingham strain, Holloway collection) and PA14 [23].

### 2.3. Heterologous Production and Purification of Dioxygenases

Recombinant *E. coli* BL21 strains were grown aerobically at 37 °C in LB with appropriate antibiotics. Cultures of *E. coli* BL21(DE3) harboring pET expression plasmids were supplemented with 0.25 mM IPTG at an OD_600 nm_ of 0.5–0.8 and incubated for approximately 16 h at 15 °C. Cultures of *E. coli* TOP10 with recombinant pBAD plasmids were supplemented at an OD_600 nm_ of 0.6–0.9 with L-arabinose (final concentration of 0.2% (*w*/*v*)) and incubated for 5–6 h at 24 °C. Cells were harvested by centrifugation, resuspended in lysis buffer (300 mM NaCl, 20 mM Tris, 0.05% NP-40, pH 8.0), and disrupted by sonication. Cell debris was removed by centrifugation (18,000 rpm, 40 min, 4 °C) and the supernatant was filtered. Proteins were purified to electrophoretic homogeneity (Appendix A, Appendix A) by Ni-NTA affinity chromatography (elution buffer: 20 mM Tris/HCl, pH 8, 300 mM NaCl, 300 mM imidazole) and stored in a buffer containing 20 mM Tris and 10% glycerol (pH 8.0) at −80 °C until required. Quantification of proteins was performed using the Bradford method, with bovine serum albumin as the standard protein, or spectrophotometrically using the NanoPhotometer^®^ N60 (IMPLEN, Inc., Westlake Village, CA, USA).

### 2.4. Enzyme Assay

The catalytic activity of the proteins was assessed spectrophotometrically by measuring PQS consumption at 337 nm and 30 °C. The assays contained up to 20 µM PQS in buffer (50 mM Tris, 2 mM EDTA, 10% PEG 1500, pH 8.0). At the specified conditions, the extinction coefficient of PQS is 10,169 M^−1^cm^−1^. One unit of enzyme activity was defined as the amount of enzyme converting 1 µmol of PQS per minute under the conditions of this assay.

### 2.5. Cultivation of P. aeruginosa in the Presence of Quorum Quenching Enzymes

Overnight cultures of *P. aeruginosa* PAO1 and PA14 in LB medium were diluted to an OD_600_ of 0.05 in 30 mL of fresh medium and incubated at 37 °C under vigorous shaking (140 rpm) throughout the whole experiment. Each of the enzymes (0.3 U/mL) was added to the growing cultures after 2 h of incubation. *P. aeruginosa* cultures supplemented with the equivalent amount of buffer (20 mM Tris buffer (pH 8), 10% (*v*/*v*) glycerol) were used as a control. To determine the production of AQs and virulence factors, culture samples were taken 6 h and 24 h after enzyme or buffer addition.

### 2.6. Determination of AQ Levels

AQ levels in *P. aeruginosa* cultures were determined with a Hitachi EZchrom Elite HPLC system. Culture samples (500 µL each) were extracted three times with 500 µL ethyl acetate (acidified with 1 mL L^−1^ acetic acid), dried to completion, re-dissolved in acidified (0.1% (*w*/*v*) citric acid) methanol, and separated by HPLC on a Eurospher II 100-5 C18 column (Knauer) at 35 °C using a linear gradient (within 40 min) from 60% methanol in water with 0.1% (*w*/*v*) citric acid to 100% methanol with 0.1% (*w*/*v*) citric acid at a flow rate of 0.5 mL min^−1^. A diode array detector (L-2450 LaChrome Elite, Merck Hitachi) was used to record the light absorption spectra of eluted compounds. PQS, HHQ, and HQNO were used as reference compounds for calibration.

### 2.7. Quantification of Virulence Factors

Pyocyanin content was quantified spectrophotometrically at 520 nm after chloroform extraction from the culture supernatant samples, as described by Essar et al. [25]. Pyoverdine levels in culture supernatants were determined by measuring the absorption at 405 nm [26]. The orcinol method, as described by Wilhelm et al. [27] was used for rhamnolipid quantification. Elastolytic activity in culture supernatants was analyzed with the elastin Congo Red assay [28].

### 2.8. Galleria mellonella Infection Model

*G. mellonella* research grade larvae (TruLarv^TM^, Biosystems Technology Ltd., Crediton, Devon, EX17 3LF, UK) were stored in the dark at 15 °C. Overnight cultures of *P. aeruginosa* PAO1 were diluted in fresh LB medium to an OD_600_ of 0.02, and bacterial cultures were grown at 37 °C under vigorous shaking. Cells were washed with sterile PBS and diluted to 10^3^ CFU/mL. Afterwards, the enzymes of interest (20 units/mL), PqsD inhibitors (100 µM), or the equivalent amount of PBS were added to 950 µL of bacteria and incubated at 30 °C for 1 h. Ten microliters of the suspension were then injected into the last proleg of a larva. Non-treated larvae were included as a negative control. In addition, larvae were injected with PBS to assess the physical trauma produced during the injection process. Following injection, *G. mellonella* larvae were incubated at 30 °C and examined every 8 h. Larvae were considered dead when not reacting to touch.

### 2.9. Statistical Analysis

Statistical differences were analyzed using GraphPad Prism Version 6.0 (GraphPad Software Inc., La Jolla, CA, USA).

## 3. Results

### 3.1. Catalytic Half-Lives of PQS Dioxygenases

To compare the stability of the newly identified dioxygenases HQD*_N.f._* and HQD*_S.b._* to that of wild-type and engineered AqdC variants, purified proteins were incubated at 37 °C and residual activities were determined. The enzymatic activity of both AqdC and HQD*_S.b._* dropped below 30% after 1 h of incubation, whereas HQD*_N.f._* and the engineered variant AqdC^V^ retained more than 80% of activity after this time interval (Figure 2). Short half-lives of the enzymes are a limiting factor for their potential application as QQ agents. Remarkably, HQD*_N.f._* combines high affinity (as suggested by a K_M_ value of 3.1 ± 0.4 µM) and high catalytic efficiency toward PQS (Appendix A) with high stability (Figure 2). The AqdC^V^ protein showed an even longer half-life (Figure 2); however, its comparatively high K_M_ value of 22.1 ± 4.1 µM (Appendix A) may limit its applicability as a QQ enzyme. HQD*_S.b._*, which exhibits the lowest K_M_ towards PQS among the enzymes tested (Appendix A), seems to be very thermosensitive in PBS, as reflected by its short half-life.

### 3.2. PQS Dioxygenases in Contrast to HodC Are Not Inactivated by P. aeruginosa Exoproducts

*P. aeruginosa* is a prolific producer of several extracellular enzymes with proteolytic activity, with the metalloprotease LasB as the most abundant protease. Previous studies have shown that HodC is rapidly inactivated in *P. aeruginosa* cultures or culture supernatants, and LasB was identified to be responsible for degradation of this enzyme [24]. Interestingly, all PQS dioxygenases tested resisted inactivation by culture supernatants of *P. aeruginosa* PA14 (Figure 3). Remarkably, even a stabilizing effect by the supernatant of strain PA14 was observed for HQD*_S.b._*, which in PBS (Figure 2) and LB medium (Figure 3) undergoes rapid thermal inactivation. Combined with its favorable kinetic properties (Appendix A), this feature makes HQD*_S.b._* a still attractive candidate enzyme for QQ assays.

### 3.3. PQS Dioxygenases from N. farcinica and S. bingchenggensis Quench AQ and Virulence Factor Production

In order to compare the quenching effects of different PQS dioxygenases on the production of AQs and selected virulence factors by *P. aeruginosa*, the same amount of enzyme units was added to growing cultures of *P. aeruginosa* PAO1 or PA14. The addition of the enzymes did not affect bacterial growth (data not shown). Besides the more thermostable AqdC variants AqdC^V^ and AqdC^VIII^, HQD*_N.f._* and HQD*_S.b._* were chosen because of their low K_M_ values for PQS (Appendix A), combined with intrinsic thermostability (HQD*_N.f._*) or resilience in *P. aeruginosa* culture supernatant (HQD*_S.b._*). All PQS dioxygenases very effectively quenched PQS levels, as indicated by the data collected 6 and 24 h after enzyme addition to the cultures (Figure 4a,b). Considering the comparatively high K_M_ value and poor catalytic efficiency of AqdC^V^ (Appendix A), it is remarkable that PQS was fully depleted in the respective samples. Levels of HHQ and HQNO were also significantly reduced in all cultures, likely due to a decrease in the expression of the AQ biosynthetic operon *pqsABCDE* under deficiency of the major autoinducer PQS. In the PAO1 cultures, HQD*_N.f._* and HQD*_S.b._* were more efficient in attenuating HHQ production than the engineered AqdC variants. Addition of PQS dioxygenases to PAO1 and PA14 cultures quenched pyocyanin and pyoverdine levels, with HQD*_N.f._* and HQD*_S.b._* eliciting the strongest attenuation in PAO1 cultures (Figure 4c,d). The PQS dioxygenases only weakly (if at all) affected the levels of elastase and rhamnolipids; however, decreases in response to HQD*_S.b._* and also HQD*_N.f._* were observed in some samples (Figure 4c,d).

### 3.4. PQS Dioxygenase from S. bingchenggensis, as Well as Small Molecule Inhibitors of AQ Biosynthesis, Enhance G. mellonella Survival upon Infection with P. aeruginosa PAO1

To test whether exogenous addition of PQS dioxygenases is able to affect the pathogenicity of *P. aeruginosa* in an in vivo infection model, we analyzed the mortality of *G. mellonella* larvae in response to injection of *P. aeruginosa* suspensions that were incubated in enzyme solution. As shown in Figure 5a, treatment of *P. aeruginosa* PAO1 with the PQS dioxygenase HQD*_S.b._* prior to injection into the larvae led to an increase (*p* = 0.0443) in overall survival as compared with the *P. aeruginosa*-infected control. Incubation of the pathogen with HQD*_N.f._* also seemed to result in a delay in larval death; however, in this case, the difference reported was not statistically significant (*p* = 0.1436). AqdC and AqdC^VIII^ did not reduce the mortality of *G. mellonella* in response to *P. aeruginosa* infection, and *P. aeruginosa*-AqdC^V^ suspensions were even more detrimental for the survival of *G. mellonella* larvae than *P. aeruginosa* in PBS. Due to its low catalytic activity, a considerably higher amount of AqdC^V^ protein was required to adjust the enzyme units chosen for the treatment. Control experiments revealed that injection of this dose of AqdC^V^ protein into larvae induced melanization and led to 86.7% larvae survival after 72 h of incubation, while no dead larvae were observed after injecting the corresponding units of HQD*_N.f._*, HQD*_S.b._*, AqdC, or AqdC^VIII^ protein, or AqdC^V^ at a 10-fold lower dose (15 larvae per condition; Appendix A). Therefore, it seems that the effect of the *P. aeruginosa-*AqdC^V^ suspension shown in Figure 5b is due to toxicity of the protein at such a high load.

To compare the in vivo effect of PQS dioxygenases, which despite efficient cleavage of PQS only partially attenuate AQ production, to that of specific inhibitors of AQ biosynthesis, infection model experiments were also performed with PqsD inhibitors developed by R.W. Hartmann and coworkers. PqsD is an early enzyme in AQ biosynthesis, catalyzing the decarboxylative condensation of coenzyme A-activated anthranilic acid with malonyl-CoA. Compound 19 ((2-nitrophenyl)(phenyl)methanol) previously was reported to strongly reduce both HHQ and PQS levels in *P. aeruginosa* [20], and compound 62 ((2-nitrophenyl)(thiophen-3-yl)methanol) was the most potent inhibitor of cellular HHQ formation among a series of derivatives based on the (2-nitrophenyl)methanol scaffold [21]. The compounds did not affect the viability of *G. mellonella* larvae at the concentration used for the assays (Appendix A). In comparison with larvae infected with untreated *P. aeruginosa*, an increase in survival was observed for larvae infected with bacterial suspensions incubated with compound 19 (*p* = 0.04830) or compound 62 (*p* = 0.0203) (Figure 6), confirming that the inhibition of PqsD is a pertinent approach to interfere with AQ-mediated QS mechanisms.

## 4. Discussion

The emergence and spread of antibiotic-resistant *P. aeruginosa* strains, especially in healthcare-associated environments, such as intensive care units, results in a drastic reduction of the effective treatment options for this opportunistic pathogen. As a result, alternative approaches to combat *P. aeruginosa* infections are under ongoing investigation, particularly those aiming at reducing the virulence of this pathogen. In contrast to conventional antimicrobial agents that impose high selective pressures on bacteria, interfering with QS signaling targets bacterial pathogenesis rather than bacterial growth. The use of QQ enzymes that inactivate extracellular signals is considered to exert less evolutionary pressure toward the development of resistance mechanisms than the use of antimicrobials or of anti-virulence compounds that act intracellularly. The PQS signal constitutes a highly specific target to address *P. aeruginosa* because PQS signaling seems to be quite unique for this pathogen. Besides *P. aeruginosa*, the marine bacterium *Pelagibaca bermudensis* was reported to produce PQS besides other AQs [29], while other known AQ producers, such as *Burkholderia* spp., do not form PQS [30]. The detection of PQS in sputum, bronchoalveolar lavage, mucopurulent fluid, blood, and urine of *P. aeruginosa*-infected CF patients [31,32,33], as well as necrotic muscle tissue from burn wounds [34] moreover suggests that AQ-dependent QS is functional during infection.

In this study, the naturally occurring dioxygenase HQD*_N.f._* has been shown to have the highest catalytic activity among the PQS dioxygenases tested, in conjunction with high affinity toward the substrate, as well as high thermal stability, making this candidate most promising for potential applications or further improvements. HQD*_S.b._*, which is also characterized by high affinity toward PQS, is of special interest due to its stabilization by *P. aeruginosa* supernatant, an observation requiring further investigation.

Interfering with the *pqs* QS system appears to hold potential as an anti-virulence approach since PQS-cleaving dioxygenases, especially HQD*_S.b._*, as well as PqsD inhibitors, were able to attenuate virulence factor production by *P. aeruginosa* and to delay its lethal effects in *G. mellonella* larvae. However, our data also show that enzymatic degradation of the PQS signal, albeit significantly reducing HHQ and HQNO production, does not fully quench AQ biosynthesis in *P. aeruginosa*. Moreover, inactivation of the PQS signal hardly affected elastase and rhamnolipid production, which is consistent with the notion that these factors, even though the *pqs* system contributes to their regulation (reviewed in [35]), are mainly controlled by the *las* and *rhl* circuits, respectively [36]. Enzymatic PQS cleavage inactivates not only a QS signal but also removes an iron trap, suggesting that quenching of pyoverdine levels may also be governed by increased iron availability, as already indicated by the study of Tettmann et al. [24].

Besides residual virulence factors still produced by HQD-treated *P. aeruginosa*, residual levels of HHQ and especially of the cytotoxic HQNO may take part in the killing of *G. mellonella* larvae. The PqsD inhibitors, which address an early step of the AQ biosynthetic pathway, should reduce the levels of all AQ and AQ-*N*-oxide congeners to a similar extent. The reduced background of cytotoxic metabolites when *P. aeruginosa* is treated with PqsD inhibitors may contribute to their beneficial effect on the survival of *G. mellonella*.

Even though an infection may not be prevented by QQ enzymes, delaying its development might allow the infected host immune system to clear the pathogen. However, despite efficient quenching of the PQS signal itself, the effects of PQS dioxygenases on *P. aeruginosa* virulence were moderate, suggesting that for *P. aeruginosa*, which regulates its virulence via a highly complex QS network, an efficient anti-virulence therapy likely requires a combination of agents that address multiple targets.

## Figures and Tables

**Figure 1 biomolecules-12-00170-f001:**
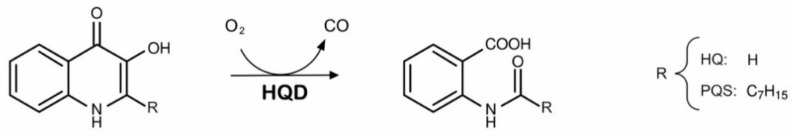
Cleavage of (2-alkyl-)3-hydroxy-4(1*H*)-quinolones as described for the subfamily of (2-alkyl-)3-hydroxy-4(1*H*)-quinolone 2,4-dioxygenases (HQDs) [14]. HQ, 3-hydroxy-4(1*H*)-quinolone; PQS, *Pseudomonas* quinolone signal.

**Figure 2 biomolecules-12-00170-f002:**
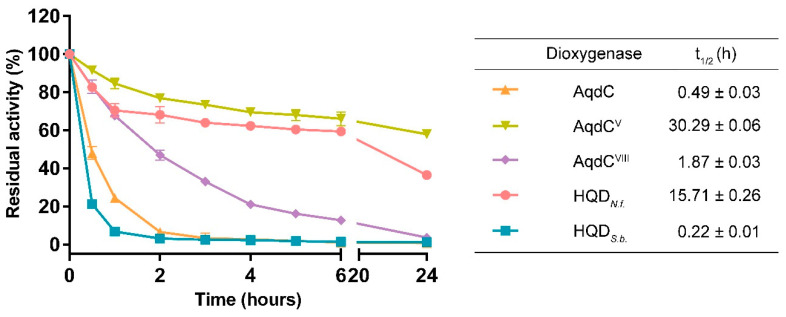
Effect of temperature on the enzymatic activity of PQS dioxygenases. Enzyme solutions (0.2 mg mL^−1^) in PBS (pH 7.5) were incubated at 37 °C, and residual catalytic activities of samples withdrawn at appropriate time intervals were measured using the standard enzyme assay. The catalytic half-life (t_½_ in [h]; period required for the enzymatic activity to decrease to half of its initial value) was determined by fitting the data to the one-phase exponential decay model, with the plateau parameter adjusted to a constant value of zero, using GraphPadPrism 6 software. Data represent the average ± SD from triplicate experiments.

**Figure 3 biomolecules-12-00170-f003:**
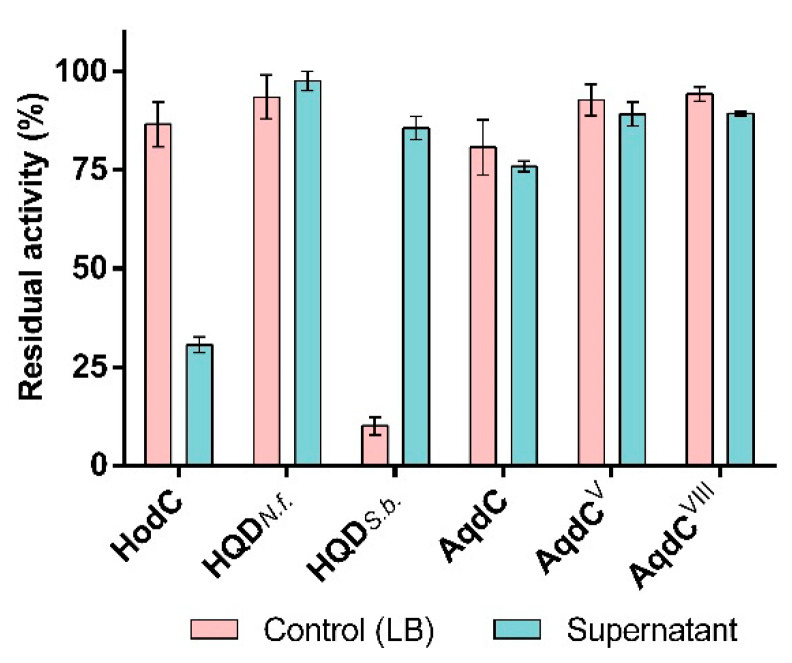
Effect of *P. aeruginosa* PA14 culture supernatant on enzymatic activity of PQS dioxygenases. HodC, which has previously been shown to be inactivated by the extracellular protease LasB, was included for comparison (see also ref. [24]). 500 µL/mL cell-free stationary phase culture supernatant of *P. aeruginosa* PA14 or 500 µL/mL LB medium (control) were incubated with the enzyme of interest (1 U/mL) in buffer (20 mM Tris, pH 8.0) at 37 °C. Residual catalytic activities after 30 min of incubation were determined in the standard enzyme assay. Data represent the mean ± SD from at least three replicates.

**Figure 4 biomolecules-12-00170-f004:**
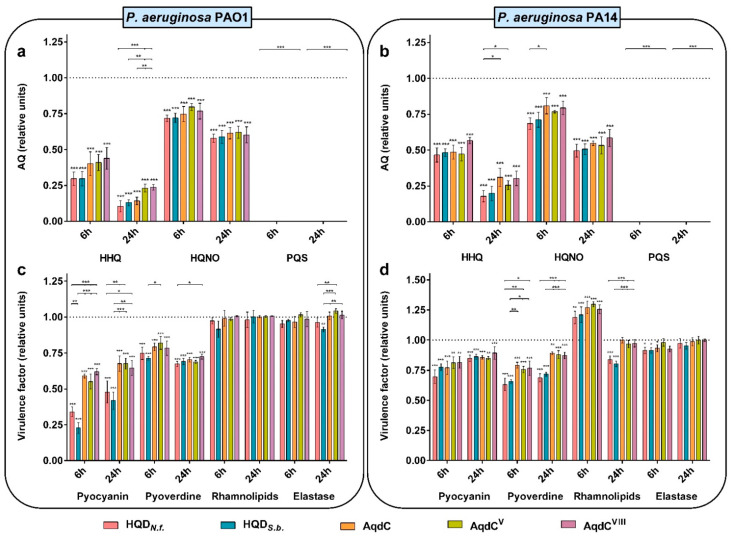
AQ levels (**a**,**b**) and virulence factor production (**c**,**d**) in growing cultures of *P. aeruginosa* PAO1 (**a**,**c**) and *P. aeruginosa* PA14 (**b**,**d**) after the addition of 0.3 U/mL of the PQS-cleaving dioxygenases HQD*_N.f._*_,_ HQD*_S.b._*, AqdC, AqdC^V^, or AqdC^VIII^. *P. aeruginosa* cultures supplemented with the equivalent amount of buffer were used as a control. Levels of AQs and virulence factors in samples of *P. aeruginosa* control cultures were set to 1 at each of the sampling times indicated (6 and 24 h) and are represented by a dotted line. Data represent the mean ± SD of three independent biological replicates. Statistical analysis was performed using ANOVA/Tukey HSD, * *p* < 0.05, ** *p* < 0.01, *** *p* < 0.001.

**Figure 5 biomolecules-12-00170-f005:**
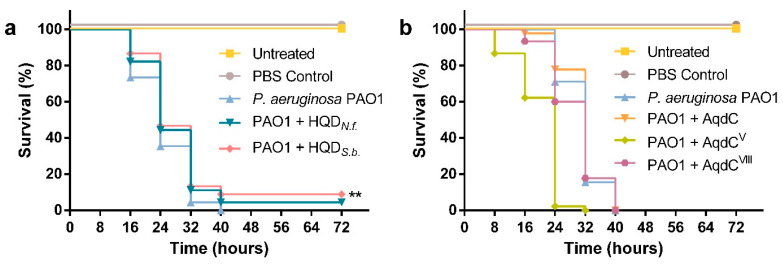
Effect of PQS dioxygenases on the survival of *G. mellonella* larvae infected with *P. aeruginosa*. Larvae were infected with 10 µL cell suspension of *P. aeruginosa* PAO1 (10^3^ CFU/mL), treated with (**a**) HQD*_N.f._* or HQD*_S.b._* (20 U/mL), or (**b**) AqdC, AqdC^VIII^ or AqdC^V^ (20 U/mL) for 1 h prior to injection of the suspension. Untreated larvae and larvae injected with the equivalent volume of PBS were included as controls. After inoculation, larvae were incubated at 30 °C in the dark and monitored every 8 h over an 80 h period. Kaplan–Meyer curves represent the results derived from three independent experiments with 15 larvae per condition (*n* = 45). ** Indicates significant differences in larvae survival in comparison with the *P. aeruginosa* PAO1 infected control (*p* < 0.05, log-rank (Mantel–Cox) test).

**Figure 6 biomolecules-12-00170-f006:**
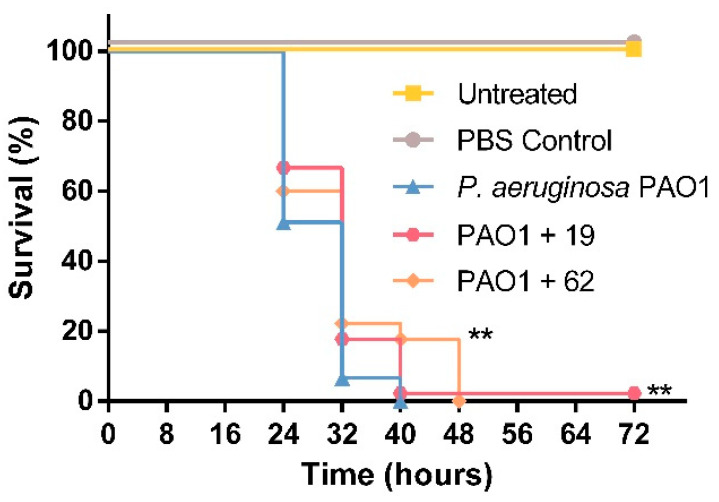
Effect of PqsD inhibitors [20,21] on the survival of *G. mellonella* larvae infected with *P. aeruginosa*. Larvae were infected with 10 µL cell suspension of *P. aeruginosa* PAO1, pre-incubated with compound 19 or 62 (100 µM) for 1 h prior to injection. Untreated larvae and larvae injected with PBS were included as controls. Post-inoculation, larvae were incubated at 30 °C in the dark and monitored every 8 h. Kaplan–Meyer curves represent the results obtained from three independent experiments with 15 larvae per condition (*n* = 45). ** Indicates significant differences in larvae survival in comparison with the *P. aeruginosa* PAO1 infected control (*p* < 0.05, log-rank (Mantel–Cox) test).

**Table 1 biomolecules-12-00170-t001:** (2-Alkyl-)3-Hydroxy-4(1*H*)-quinolone 2,4-dioxygenases (HQDs) examined in the present study. Expression plasmids and the respective source organisms of the enzymes are indicated. European Nucleotide Archive (ENA) accession IDs of the genes are also included.

Protein	Source Organism	ENA ID	Expression Plasmid	Reference
**HodC**	*Arthrobacter* sp. Rue61a	CAA75080.1	pET23a::*hodC*	[24]
**AqdC**	*Mycobacteroides abscessus* ATCC 19977	CAM60402.1	pET28b::his8-TEV-*aqdC*^I^	[17]
**AqdC^V^**	*Mycobacteroides abscessus* ATCC 19977	CAM60402.1	pBAD::his8-tev-*aqdC*- G40K-A134L-G220D-Y238W	[19]
**AqdC^VIII^**	*Mycobacteroides abscessus* ATCC 19977	CAM60402.1	pBAD::his8-tev-*aqdC*- G40K-G220D-Y238W	[19]
**HQD*_N.f._***	*Nocardia farcinica* IFM 10152	BAD60071.1	pET28b::his8-TEV- *HQD_N.f._*	[14]
**HQD*_S.b._***	*Streptomyces bingchenggensis* BCW-1	ADI11806.1	pET28b::his8-TEV- *HQD_S.b._*	[14]

## Data Availability

The data presented in this study are available in the Figures of the article and the Appendix A. Strains are available upon request from the corresponding author.

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
