# Peer review of "Enzyme-Mediated Quenching of the Pseudomonas Quinolone Signal (PQS): A Comparison between Naturally Occurring and Engineered PQS-Cleaving Dioxygenases"

_biomolecules, 2022, doi:10.3390/biom12020170_

Round 1
Reviewer 1 Report
The authors tested the ability of different Quorum sensing molecule degrading enzymes to influence Pseudomonas aeruginosa pathogenicity. The used enzymes were isolated from other bacterial species and in part engineered for better stability. Firstly, the authors showed that these enzymes are able to degrade QS signals of the AQ System of P. aeruginosa. They secondly examined the effect of these enzymes on different virulence factors known to be regulated by QS and in a Galleria mellonella infection model. The idea that to use QQ enzymes to influence bacterial signals outside the cells in an infection and therefore exert less evolutionary pressure is very promising.
General concept comments
The authors have a long record in this field and a high expertise with the PQS dioxygenases. In general, the introduction and the protein part are very well written, the experiments well-structured and the results clearly presented. The authors show the ability of the enzymes to degrade the QS signal molecules of P. aeruginosa.
The weakness of the study is in my opinion the virulence experiments and the discussion. The chosen virulence factors examined and the infection model are not appropriate to study the effect of the quenching enzymes very well. For example, in line 263-265 it is mentioned that rhamnolipid and elastase are mainly controlled by other systems, the question is why they are then chosen to study the PQS dioxygenases. Overall this is an information, that is nice to know, but does not help to prove the principle and is not further discussed. The discussion in general is partly included in the results part which in turn leads to a very short discussion.
Figure 3: For this experiment it was not very clear to me if the “singal molecules in P. aeruginosa control cultures” which were set to 1, are the concentration at time point 0 or are they also at time point 6 and 24h set to 1. It is likely that the concentration in the control culture are increasing. Is this the case? Here the experiment conception should be better described.
Very important to evaluate this experiment at all, is to know if there is an effect on the bacterial viability? Is the culture growing during the experiment? Have the enzymes an effect on the viability of P. aeruginosa (in comparison to the “control culture”. If the culture is in stationary phase many virulence factors have than a higher expression.
Figure 4: Is Missing?!
Figure 5: For the Galleria mellonella infection experiment the control with PBS and enzymes is not shown. It is mentioned in the results part, but is very short and should be include at least in the supplemental part. This toxic effect should be tested on cell cultures in future experiments.
Figure 6: For the Galleria mellonella infection experiment the control with PBS and inhibitor is not shown and not mentioned in the text. Is there an effect of the inhibitor on the bacteria directly?
Further interesting experiments (just ideas, maybe already considered): Is there an effect on antibiotic susceptibility, if the enzymes are added with an antibiotic? Is there an effect on cell cultures viabilty if the enzymes were added together with P. aeruginosa.
Specific comments :
Introduction:
Line 76 ff: A reference/literature is missing for HodC degradading PQS.
Line 102: strain PA14 is not mentioned here.
Material and Methods:
Line 109: What is PqsD? It is not described in the introduction, here is its first mention and is also in the results part very briefly described.
Results:
Numbers of the figures in the text are in many cases wrong! Line 232, 233, 234 ff
Line 220-225 This part belongs in my opinion to the discussion not to the results part.
Line 257 – 266 This part also belongs in my opinion to the discussion not to the results part.
Figure 3 legend Line 271 – 272 This belongs also to the M&M section, there it is not mentioned.
Figure 3: I would prefer to have a labeling wich parts belong to PA14 and which to PAO1 (maybe on top) not just mentioned it in the legend.
Author Response
Please see the attachment. The blue text shows the comments and suggestions of the reviewer again, the black text our responses.

Reviewer 2 Report
The authors of this study investigates the usefulmess of HQD dioxygenases in quenching the levels of the quorum sensing molecules PQS and HHQ produced by Pseudomonas aeruginosa PAO1 and PA14. They compared the efficiency of different enzymes already described or new ones from Nocardia and Streptomyces ( HQDnf and HQDsb), respectively. This study shows that these two last enzymes are active against PQS and HHQ and cause a decrease in pyocyanin and pyoverdine production in P. aeruginosa. Interestingly, HQDsb being the least stable at 37° in PBS, shows increased stablity in P. aeruginosa supernatant. Finally, the authors tested the efficacy of the QQ activity of these two enzymes in protecting Galleria melonella larvae from infection, but with disappointing low efficiency, which they acknowledge in the Discussion section.
Remarks:
- Figure 2: Could it be that the HQDsb is protected by its substrate (PQS or HHQ). I suggest to use a supernatant of a pqsA mutant to see whether the protecting effect is still observed.
- Figure 5: The enzymes are maybe affected by G. mellonella proteases?
- An alternative virulence assay on chicoree leaves couls also be performed (or other plants). Perhaps a combination with protease inhibitors could be used.
- The effect on pyoverdine production might be indirect since PQS is known to bind Fe3+ and consequently induce pyoverdine production. A pqsH mutant which fails to convert HHQ to PQS should not show an effect on pyoverdine production, since HHQ does not complex iron.
- Figure S1: two lanes are labeled HQDnf.
Author Response

(The authors gave the same response as above.)

Reviewer 3 Report
This is a thorough, very well written manuscript describing enzyme-mediated quenching of quinolone signaling in Pseudomonas, a potential new treatment option for this bacterial infection.
There are only minor typos to be correted, as the overall manuscript is excellent and scientifically sound:
l. 20: subscript 'M' in 'KM';
l. 254: remove line break;
l. 329: remove additional space in "healthcare-associated environments".
Author Response
Please see the attachment. The blue text showd the comments of the reviewer again, the black text is our response.

Round 2
Reviewer 1 Report
The authors improved the manuscript very good.
In my opinion there are still some easy exeperiments (anitbiotics + enymzmes) which would improve the impact of the manuscript. But I accept the opinion of the authors that they pick some important easy to measure virulence factors and further experiments are "beyond the scope of this manuscript"
I also agree that the Galleria model is a good model to check on virulence. My points are just that the experimental procedure was not that easy understandable and controls are missing. But these are now better described and added to the supplemental files.
Reviewer 2 Report
This new version of the manuscript and the response to reviewers are ok.